# Changing antibiotic prescribing practices in outpatient primary care settings in China: Study protocol for a health information system-based cluster-randomised crossover controlled trial

Yue Chang [1]☉*, Yuanfan Yao[2]☉, Zhezhe Cui[3]☉, Guanghong Yang[2]*, Duan Li[2]*, Lei Wang[4], Lei Tang[2]

1 School of Medicine and Health Management, Guizhou Medical University, Guiyang, Guizhou Province, China, 2 School of Public Health, Guizhou Medical University, Guiyang, Guizhou Province, China, 3 Guangxi Zhuang Autonomous Region Center for Disease Control and Prevention, Nan'ning, Guangxi Province, China, 4 Primary Health Department of Guizhou Provincial Health Commission, Guiyang, Guizhou Province, China

☉ These authors contributed equally to this work.
* 4567401@qq.com (YC); 280446859@qq.com (GY); 10958391@qq.com (DL)

**Funding:** The study was funded by the National Natural Science Foundation of China Grant on

## Abstract

### Background

The overuse and abuse of antibiotics is a major risk factor for antibiotic resistance in primary care settings of China. In this study, the effectiveness of an automatically-presented, privacy-protecting, computer information technology (IT)-based antibiotic feedback intervention will be evaluated to determine whether it can reduce antibiotic prescribing rates and unreasonable prescribing behaviours.

### Methods

We will pilot and develop a cluster-randomised, open controlled, crossover, superiority trial. A total of 320 outpatient physicians in 6 counties of Guizhou province who met the standard will be randomly divided into intervention group and control group with a primary care hospital being the unit of cluster allocation. In the intervention group, the three components of the feedback intervention included: 1. Artificial intelligence (AI)-based real-time warnings of improper antibiotic use; 2. Pop-up windows of antibiotic prescription rate ranking; 3. Distribution of educational manuals. In the control group, no form of intervention will be provided. The trial will last for 6 months and will be divided into two phases of three months each. The two groups will crossover after 3 months. The primary outcome is the 10-day antibiotic prescription rate of physicians. The secondary outcome is the rational use of antibiotic prescriptions. The acceptability and feasibility of this feedback intervention study will be evaluated using both qualitative and quantitative assessment methods.

"Research on feedback intervention mode of antibiotic prescription control in primary medical institutions based on the depth graph neural network technology" (71964009) and the Science and Technology Fund Project of Guizhou Provincial Health Commission Grant on "Application Research of Deep Learning Technology in Rational Evaluation and Intervention of Antibiotic Prescription" (gzwjkj2019-1-218). The funders had a role in the study which we should acknowledge. Specifically, all funders provided travel expenses during the data collection process, as well as the expert's expenses for providing guidance on the study design, technological guidance and data analysis.

**Competing interests:** All authors have no conflicts of interest to declare.

**Abbreviations:** IT, information technology; AI, artificial intelligence; DMNN, depth map neural network; HIS, Health Information System; LWTC, Lianke Weixin Technology Co., Ltd; ICGPHC, Guizhou Provincial Health Commission Information Center; GNN, graph neural network technology; ICD-10, International Classification of Diseases 10th Edition; CG, control group; IG, intervention group; ITT, intention-to-treat; CONSORT, Consolidated Standards of Reporting Trials.

## Discussion

This study will overcome limitations of our previous study, which only focused on reducing antibiotic prescription rates. AI techniques and an educational intervention will be used in this study to effectively reduce antibiotic prescription rates and antibiotic irregularities. This study will also provide new ideas and approaches for further research in this area.

## Trial registration

ISRCTN, ID: ISRCTN13817256. Registered on 11 January 2020.

## Background

Antibiotic resistance is a widespread concern around the world [1]. In the past decade, 50% of antibiotic prescriptions worldwide have been used to treat colds, and, according to a report of World Health Organization (WHO), many of these cases have no indication of antibiotic use [2]. The main forms of antibiotic overuse and abuse are non-indication, overdose, and multi-drug combined use of antibiotics which violate the principles of antibiotic use [3, 4]. Overuse and abuse of antibiotics are major risk factors for antibiotic resistance [5–9]. The total consumption of antibiotics in 71 countries (including China) increased by more than 50% from 2000 to 2010 [10]. The antibiotic prescription rate in China is twice as high as that recommended by WHO, and higher than that in developed countries and most developing countries [9–11]. Guizhou province, located in southwest China, has the largest number of poor people and is the largest poverty-stricken area in China. Our retrospective study of Guizhou's 39 primary care facilities showed that most of the medical staff have less than a university education, and the unreasonable rate of antibiotic prescriptions was over 90%. The incidence of bacterial resistance in Guizhou province is on the rise [12–14].

Previous studies have shown that there are a variety of methods for antibiotics prescription control, including educational intervention [15], communication training [16], nursing point testing [13], electronic decision support system [17], and delayed prescription [18], but reports of a feedback intervention are rare. Existing feedback interventions mainly focused on email or poster information [18–26], regular or irregular assessment/audit of antibiotic prescriptions [21, 22, 25–29], or prescription recommendations from experts and peers delivered at a meeting or online [18, 19, 24, 25, 30–32]. Some studies even report prescribing information publicly [20, 33]. However, these methods are somewhat mandatory and censored, which can cause negative emotions to the physician, and needs long-term intervention by professionals [34–36]. As a result, some studies have shown negative results [37, 38].

We previously conducted a cluster randomised crossover-controlled trial to reduce antibiotic prescription rates based on existing health information system (HIS) in primary care institutions in Guizhou [39]. In this study, the antibiotic prescription rates of the two groups decreased by 15% over the 6-month study period. However, a limitation was that prescription of unreasonable antibiotics were not considered.

In view of this, this new study will analyze and process the digital information in electronic medical records with big data technology, and use the depth map neural network (DMNN) in artificial intelligence (AI) technology to provide physicians with the best diagnosis and treatment suggestions in real time.

This is a cluster-randomised, open controlled, crossover, superiority trial. We aim to describe an automatically-presented, privacy-protecting, DMNN technology-based feedback intervention model. The feedback intervention model can not only effectively remind physicians of the deviation of their prescribing behavior, but also humanely give reasonable suggestions, which can greatly improve the enthusiasm of physicians to participate. In addition, an educational handbook developed by us for primary outpatient institutions will be distributed to primary physicians. A pilot study will be conducted to test the physician motivation and intervention effectiveness. The comparators are usual care i.e., primary care hospitals within Guizhou which did not receive any intervention.

## Methods design

### Trial setting

The trial will be carried out in primary care institutions in four geographical regions of Guizhou province: the east, west, north, and south. We have identified the primary care institutions as township public hospitals and community health service centers in a previous study, which provide primary health care services to the majority of rural residents in China [39]. Guizhou province has a population of approximately 39 million and is one of the most impoverished provinces in China. A township public hospital or community health service center is a comprehensive institution for health administration and medical prevention work established by a county or township. Up to 2019, there were 1,329 township hospitals in Guizhou province, with only 7,211 practicing physicians, most of whom have only received vocational education equivalent to a junior college or technical secondary school level [39–41]. According to our 2018 study of 16 primary hospitals in rural areas of Guizhou province, most (63%) of the antibiotic prescriptions were made by resident physicians with a below college level of education and most antibiotic prescriptions were deemed to be inappropriate [13].

The Health Information System (HIS) involved in this study was designed and developed by Guizhou Lianke Weixin Technology Co., Ltd. (LWTC) under the authorization of Information Center Guizhou Provincial Health Commission (ICGPHC). By accessing the port of ICGPHC platform, relevant data can be obtained. All of the research team's preliminary research data will come from the platform. One of the interventions included in this study, the antibiotic prescription rate ranking feedback intervention early warning system, was jointly developed by the company's technical staff based on the platform and the applicant's study requirements. It has been successfully implemented for 6 months from February to August 2018 in 31 primary care settings in Guizhou province [39].

**Graph neural network technology.** Graph neural network technology (GNN) is an advanced form of AI technology [42]. A GNN model can realize the formulation and recommendation of an ideal treatment plan according to the optimized causal reasoning function of the model, develop an AI real-time warning system for unreasonable antibiotic prescription, and conduct intelligent and reasonable interventions on the antibiotic prescription patterns of physicians in primary care institutions. In this process, the network system will involve repeated self-learning and correction to improve the early warning system for the unreasonable use of antibiotics [43–45]. Finally, through the evaluation of the intervention effect of the multi-level model, an intervention model of antibiotic prescription can be obtained to provide an economic, feasible, and effective reference plan for solving the overuse and abuse of antibiotic prescription in primary care institutions, thus reducing the drug resistance rate and burden of public finances in rural areas of China.

## Recruitment

**Clusters.** According to the inclusion criteria of the previous study [39], a cluster is defined as one in which: 1) the primary care institutions are in Guizhou province and have the same HIS system; 2) each primary care institution has at least 3 outpatient general physicians (GPs), each of whom has a history of seeing at least 100 patients, on average, every 10 days; 3) all physicians have worked in the hospital in their current position for more than 1 year. In 2020, 252 primary care institutions in Guizhou used the LWTC HIS system of which 132 met the eligibility criteria. These institutions were randomly located in 6 cities of Guizhou province—Bijie, Zunyi, Tongren, Anshun, Qiannan and Liupanshui.

**Patients.** The study subjects were primary care institutions that provided health care services for township residents in Guizhou province. For this purpose, identified eligible patients as all patients who received an initial diagnosis and were prescribed by an outpatient physician at the participating care institutions. The main diagnostic categories for all diseases were based on the International Classification of Diseases, 10th Edition (ICD-10) codes [46].

## Process

A complete trial process is shown in Fig 1. To avoid selection bias [47] we will use a crossover design in which each group will receive different treatments at different times [48]. We will divide outpatient physicians in each selected primary care institution equally into two sequences according to the principle of randomization.

The trial will be divided into two stages and will last for 6 months. The first stage will last for 3 months, with the Group 1 enrolled in the intervention group and the Group 2 in the control group. The second stage will also last for 3 months, with the two groups crossing over. The cross-over design is shown in Fig 2.

**Control.** In the control group, no form of intervention will be provided. The control group of physicians participating in the trial will continue to provide their usual treatment methods and experience to diagnose and treat patients.

**Intervention.** In the intervention group, based on previous trials [39], an antibiotic feedback intervention composed of three parts will be developed, including a real-time warning of improper antibiotic use and a 10-day summary of antibiotic prescription rate ranking and related information. The distribution of homemade educational manuals will also be made (details to be given in section 3).

*(1) AI-based real-time warning pop-up windows of improper antibiotic use.* Based on the HIS system of primary care institutions, the warning plug-in uses graph neural network technology to automatically access the prescription data in the background. It will compare each prescription with the big data and DMNN modeling results, determine whether the antibiotic prescription (including type, dosage, and course of treatment) is reasonable to be used in the consultation service and will provide a real-time automatic warning alert for unreasonable antibiotic prescription. Once a physician prescribes an unreasonable antibiotic, a pop-up window will automatically appear in the lower right corner of the screen to alert the physician that the prescription is unreasonable and indicate the type of unreasonable use of antibiotics. The form of pop-up window is shown in Fig 3. The pop-up window will disappear if the physician clicks on it. It will also automatically disappear after 5 minutes. The duration of the pop-up window will be recorded automatically by system. Extreme durations will be noticed (i.e., 1 second or 5 minutes). According to previous research [13], we define unreasonable prescription of antibiotics with the following indicators: 1. Incorrect or unnecessary use: for example, a physician gives antibiotics for which there is no clear

| | STUDY PERIOD | | | | |
|---|---|---|---|---|---|
| | **Enrolment** | **Allocation** | **Post-allocation** | | **Close-out** |
| **TIMEPOINT(Month)** | **-T1** | **T0** | **T1** | *T2* | **T3** |
| **ENROLMENT:** | **1** | **2-3** | **4-6** | **7-9** | |
| **Eligibility screen** | X | X | | | |
| **Informed consent** | | X | | | |
| **Randomised Allocation** | | X | | | |
| **INTERVENTIONS:** *Intervention Group (IG)* *Control Group (CG)* | *1:The AI-based real-time warning of improper antibiotic use* *2:The pop-ups of antibiotic prescription rate ranking* *3:Distribution of educational manuals* | *Group 1* *Group 2* | *IG1 ⇨ CG1* *CG2 ⇨ IG2* | | |
| **ASSESSMENTS:** *Antibiotic Prescription rate* | | X | X | X | X |
| *Rational rate of Antibiotic Prescription* | | X | X | X | X |
| *Covariate (Information of physicians and patients)* | | X | | | |

**Fig 1. Overview of enrollment, intervention, and assessments of the cross-over design trial.**

indication; 2. Incorrect antibacterial spectrum: for example, prescribing aminoglycoside drugs for gram-positive bacteria; 3. Combined antibiotic use: administration of more than one injectable or oral system antibiotic at a time without any indication, for example, amoxicillin capsule and ceftazidime injection in combination.

*(2) Pop-up windows of antibiotic prescription rate ranking.* This reminder system is a plug-in developed in a previous study [39]. We will implement pop-up windows of antibiotic prescription use in the HIS system. The system will appear on the physician's screen in the form of an automatic pop-up window every 10 days, informing them of their ranking in

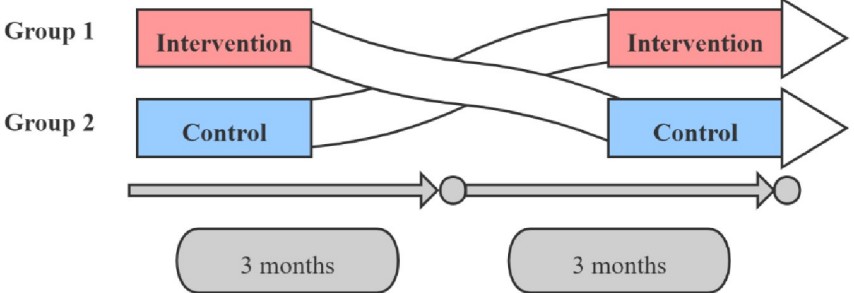

**Fig 2. Cross-over trial diagram.**

terms of their antibiotics prescription rate within the same outpatient department, actual antibiotic prescription rate and related information. The information seen by each physician will be confidential. The physicians have the freedom to read this feedback massage or not. When the physician logs into the HIS, a pop-up window or link will appear on the computer screen, prompting him or her to view the message. If a physician presses the ESC button, it will disappear. All the on-screen procedures, including click rate and the time of the message, will be recorded automatically.

Based on a previous study involving 16 hospitals in the early stage [13], we will invite 48 medical experts to conduct two rounds of demonstration using the Delphi method. Two guidance proposals with expert consensus will be formed as detailed in the section below.

*(3) Distribution of educational manuals.* The educational manuals include 2 parts: "Instruction and Recommendations for Outpatient Clinical Use of Antibiotics in Primary Care

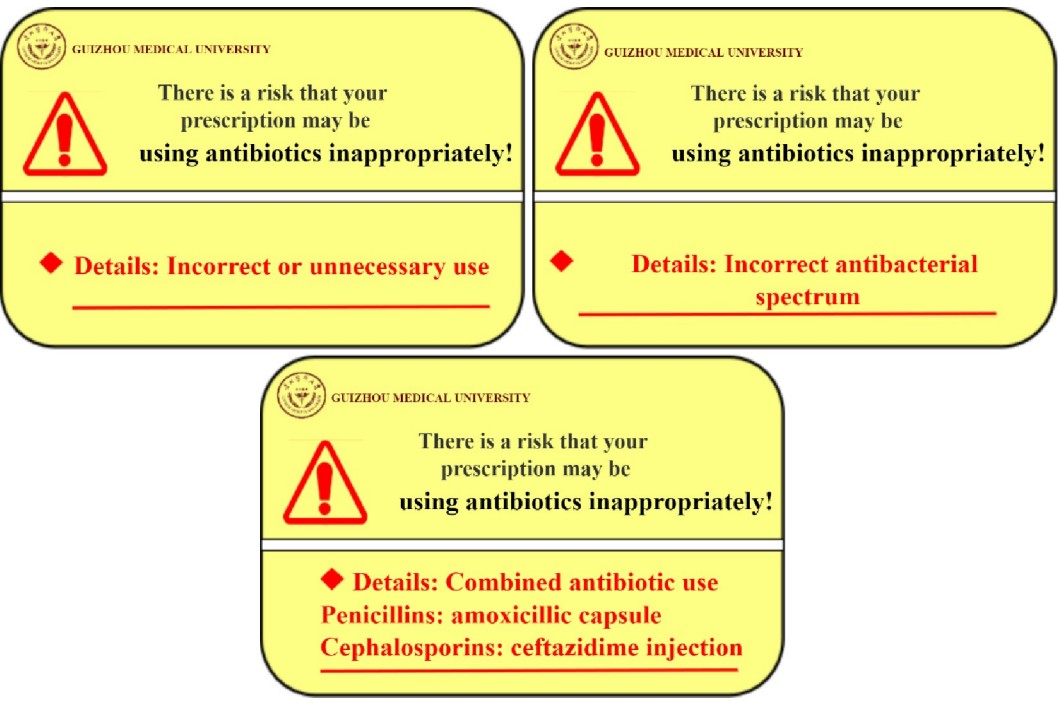

**Fig 3. Example of unreasonable use of antibiotic warning pop-ups.**

Institutions" and "Instruction and Recommendations for Diagnosis of Common Infectious Diseases in Outpatients of Primary Care Institutions". We have consolidated them into a manual for distribution to outpatient physicians in primary care institutions.

The first part is the recommendation for the rational use of antibiotics, and the second part is the diagnostic guidance of the symptoms, signs and auxiliary examinations for common infectious diseases such as digestive system, respiratory system, and urinary system.

In the first part, we divided the criteria for rationality of antibiotics into four categories: 1. Suitable: preferred antibiotic; 2. Optional: the antibiotic can be used or substituted; 3. Wrong-spectrum: the antibacterial spectrum is not used correctly; 4. No use: In the second part, based on the proportion of different diagnostic criteria and the weight, we set the most valuable diagnostic criteria as "4", and the standard for low diagnostic value was set to "1".

## Pilot study

Prior to the formal trial, we will conduct a pilot study to test the feasibility of this intervention trial. Specifically, we will work with HIS engineers to test the sensitivity and reliability of our newly developed AI-based early warning system, which is based on the existing pop-ups of antibiotic prescription information. We will also distribute the educational materials to physicians. Our study group members will be trained to explain the intervention process to the relevant manager and physicians.

The pilot study will be conducted in the outpatient department of a hospital. We will interview the director of the hospital and all qualified outpatient physicians. They will be informed about the precautions, processes, risks and benefits of the pilot study and the method of data collection in the informed consent form. We will include all physicians who signed the informed consent form in accordance with the intention-to-treat (ITT) principle [49]. And we will follow the same outline as in the formal trial.

The pilot study will last 3 months. At the end of the pilot study, the attitude, opinions, and suggestions of physicians on the feedback intervention will be obtained through a questionnaire survey. According to the feedback results of the questionnaires and the field work conducted during the pilot study, we will determine the following research points before the formal intervention trial: 1. Whether the newly developed AI-based plug-in can realize long-term, large-scale and high-precision real-time warning of prescriptions; 2. Whether all the pop-up windows and links of the warning system can work normally; 3. Whether physicians can understand and grasp all the antibiotic prescription intervention information reasonably quickly; 4. Whether the feasibility of the feedback intervention, specifically, the majority of primary care institutions and their outpatient physicians think that our research work is feasible. In other words, most of the outpatient physicians who received the intervention will feel that our study is helpful to their work by the end of the trial and will continue to use our feedback intervention.

## Data collection and management

Approved by ICGPHC, with the help of engineers from LWTC, we will retrieve antibiotic prescriptions and total prescriptions of the HIS in primary care institutions for statistical analysis through the downloaded program written by engineers. Data collection will proceed from April to October 2021. All data will be collected real-time from ICGPHC's data center, thus there will be no interference to hospitals and physicians during the data collection process. All researchers participating in the data collection will sign a confidentiality agreement.

Due to the large amount of data, the downloading process and management of prescription data will be the responsibility of the two main authors (YYF and CY). The collected data will

be entered into a standard format database to store all the outcome and covariate data. At the same time, we will generate codes that connect physicians and patients to facilitate the analysis and processing of individual-level data. In addition, the demographic and professional information of physicians will be obtained from the personnel management department of the primary hospital.

## Randomization and blinding

After recruitment, the information technology staff of LWTC will be invited to randomly assign (computer-generated random numbers) all recruited primary care institutions to an intervention group and a control group in a ratio of 1:1.

Since this is a behavioural intervention trial, the physicians will have a clear idea of the intervention when they sign the informed consent form and will be able to determine whether they are in the intervention or control group at the start of the trial. Therefore, the design of this study makes it impossible to use a blinded approach to participants and researchers [39].

## Outcome measures

**Primary outcome.** The primary outcome is the 10-day antibiotic prescription rate of physicians defined as the number of antibiotic prescriptions divided by the total number of prescriptions in each 10-day time period (the term "prescription" as used here refers to one drug) [39]. Indicators of related covariates include: 1. Baseline characteristics of the physician, such as age, gender, job title, education, and working years; 2. Patient characteristics, such as gender, age, ethnicity, and disease; 3. Antibiotic information, such as name, dosage form, route, and amount in grams.

**Secondary outcome.** The secondary outcome is the rational rate of antibiotic prescription defined as the reasonable amount of antibiotics prescribed during the study period divided by the total amount of antibiotics prescribed though the study period.

**Sample size calculation.** Since the outcome variable (antibiotic prescription rate per physician) is a continuous variable, we used the two independent means formula (two-tailed test) to calculate the required number of physicians to recruit into the study as given by the following formula.

$$n_1 = \frac{\left(z_{1-\frac{\alpha}{2}} + z_{1-\beta}\right)^2 \left[\sigma_1^2 + \frac{\sigma_2^2}{r}\right]}{\Delta^2}$$

$$r = \frac{n_2}{n_1}, \Delta = \mu_1 - \mu_2$$

In the above formula, the parameters are as follows: $\alpha = 0.05$, $Z(0.975) = 1.96$, $\beta = 0.2$, $Z(0.8) = 0.84$. Based on the data from our previous study, the pre-intervention mean ($\mu1$) = 35.0, and the pre-intervention variance ($\sigma1$) = 15.0; the post-intervention mean ($\mu2$) = 30.0, and the post-intervention variance ($\sigma2$) = 15.0 [39]. Since a 1:1 ratio was adopted in the experimental design, the sample size ratio (r) of the two groups was 1.0. From this we can calculate the sample size as n1 (group 1) = n2 (group 2) = 142 physicians per group.

To allow for a 10% non-response rate, the sample size was increased to 160 physicians per group for a total of 320 physicians. Since most of these primary hospitals have 3–4 outpatient physicians who meet the inclusion criteria, we will randomly select hospitals using a computer-generated number from the list of 132 hospitals that met the inclusion criteria. The total number of hospitals to be included in the study will be determined by whether the hospitals

have 320 outpatient physicians who meet the inclusion criteria. We will visit the selected hospitals and ask the physicians to sign the informed consent form. Fig 4 shows a flow chart of the trial where group 1 represents the group of physicians who will receive the intervention in the first stage.

## Statistical analyses

We will follow the statistical method of the previous study [39]. Descriptive statistics will be presented for the outcome variables and related covariates. We will compare the antibiotic prescription rate and the antibiotic prescription rationality rate between the two groups (Group 1 and Group 2) at baseline, crossover point and at the end of the trial. After testing of the data for normality, Student's t-test, paired t-test, Wilcoxon signed-rank test or Rank-sum test will be used to compare antibiotic prescription rate and antibiotic rationality rate between the two groups for horizontal (between groups) comparison or vertical (at different time points within the same group) comparison, to observe the difference and trend of change. Secondly, because it will take some time before feedback interventions have any impact on antibiotic prescription rates, a transition model suitable for studies that includes regular follow-up intervals and changing exposure and outcome states will be used to predict the impact of the intervention on changes in antibiotic prescription rates and rationalization rates within the same physician

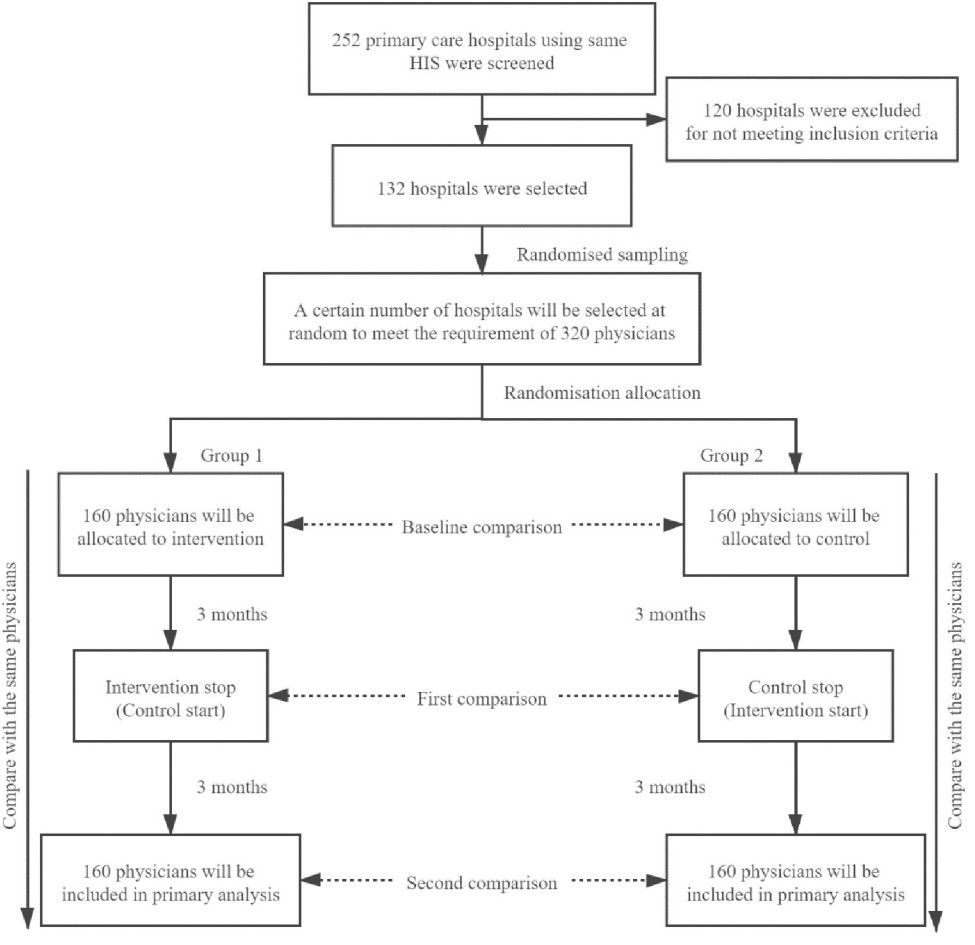

**Fig 4. Flow chart of the crossover trial.**

over time. The transition model ensures that the "carry-over" impact is adjusted and that correlations over time are addressed.

Following ITT principles [49], outpatient physicians from all participating primary care facilities will be included in the analysis. We will report all results according to the CONSORT guidelines [50].

All data analyses will be done using R version 4.0.4

## Process evaluation

Based on the Medical Research Council's 2008 framework [51] and Grant's framework for process evaluations of cluster randomised trials of complex interventions [52], we will carry out a mix of qualitative and quantitative process evaluation method.

The purpose of the process evaluation is to determine if our complex interventions are effective. Therefore, we will focus on the following key research questions that need to be addressed: "What part of the feedback intervention worked?", "Is the intervention effective for the target population?" and "Why are feedback interventions effective?" to set up the process evaluation plan. The specific evaluation content of the evaluation will include the following aspects: 1. To evaluate the feasibility, universality and acceptability of the AI-based real-time warning system of inappropriate use of antibiotics, which is based on GNN technology, and to assess the warning system for a high proportion of antibiotic prescription; 2. To assess the feasibility and reliability of the educational materials; 3. To evaluate the sampling and recruitment process at the cluster level (primary care hospitals) and individual level (outpatient physicians); and 4. To assess the response of the study to the intervention trial at the cluster level and the individual level.

We will use document review (recruitment standards, informed consent, and education manual), telephone return visit of outpatient physicians in primary care institutions and questionnaire survey of intervention trials as the evaluation methods of the process evaluation. The survey and interview guidelines used in the process evaluation are based on the Theoretical Domains Framework guidance [53]. This will give us a deeper understanding of how feedback interventions work. We will use the same method for data collection in both the control group and the intervention group. The prescription data will be aggregated every 10 days.

For the qualitative study, the sample size will be determined based on the feedback intervention test results. The sample size will be adjusted midway during the study according to the personnel turnover and the situation of withdrawal from the study. The qualitative research method will adopt the explanatory description method.

The results of the process evaluation will provide useful information for future feedback intervention trials.

## Trial management

Prof. Yue Chang and Mr. Yuanfan Yao from the Guizhou Medical University and Dr. Zhezhe Cui from the Guangxi Zhuang Autonomous Region Center for Disease Control and Prevention will be the co-guarantees of the trial and will have full access to the trial dataset. We have signed a data confidentiality agreement with the Guizhou Provincial Health Commission to protect the safety and privacy of the physicians, patients concerned and to ensure that all data is collected in accordance with accepted ethical guidelines, properly stored and used for research purposes only. We will also invite experts in relevant fields to set up a trial steering committee. Conference calls will be held every month until the completion of the study. Committees may also meet on an ad-hoc basis as required. Any modification of the agreement will be decided by the meeting.

### Trial registration

The trial was registered at **Current Controlled Trials: ISTRCTN13817256** on **11 January 2020**.

### Trial status

We expect to formally launch the trial in April 2021. The pilot study has started under the consent of the local health department and will be finished by the time of this proposal submission. We aim to publish the results in international medical journals, and promote our research results in Guizhou province.

### Ethical approval and consent

The trial has been approved by **Human Trial Ethics Committee of Guizhou Medical University (Certificate No.: 2019 (148)) in Dec. 27, 2019**. All the physicians at the primary care institutions participating in the trial will sign the informed consent. When collecting and analyzing data, we will remove patient confidential information, such as name and national ID. Physicians' prescription data will also be subject to strict confidentiality measures.

## Discussion

This study builds on a previous feedback intervention trial conducted by our research team in primary care institutions in Guizhou province [39], where we conducted the trial based on HIS and successfully reduced antibiotic prescription rates among outpatient physicians. Our current study has some advantages over this and other previous studies.

The purpose of this intervention is to establish a highly compliant, economic, and feasible artificial intelligence early warning system for antibiotic prescription control in primary care institutions of Guizhou province. The system adds an AI-based real-time warning of inappropriate antibiotic use and self-compiled education manuals to a previous study [39] that included only a 10-day pop-up alert message of antibiotic prescribing rate ranking and related information. Based on the results of our preliminary study, this intervention is suitable for primary care institutions in developing countries with paperless office conditions.

To overcome the limitations of our previous study, which only focused on reducing antibiotic prescription rates, new techniques and evaluation indexes will be used to expand our study. GNN technology will be adopted in this study to develop an AI real-time warning system for unreasonable antibiotic prescriptions, to make intelligent judgments and interventions on antibiotics prescribed by the primary care physicians. This new heterogeneous and composite network structure model and iterative optimization method will make the GNN not only have the performance of traditional high network expression ability, but also avoid the problems of traditional deep learning technology, such as complex iteration, poor understanding, and poor interpretation [44, 45]. Once the research results are applied into practice, they will provide more effective, convenient, quick, and economical intervention measures for preventing the overuse and abuse of antibiotics in primary care institutions.

In addition, we will invite 48 domestic experts to construct a recommended manual for rational use of antibiotics and diagnosis of common infectious diseases for primary care institutions using the Delphi method, which we expect to be highly praised by physicians and hospital managers in the pilot study and will have a good guiding effect on primary care institutions [54, 55].

Despite these advances, our study will inevitably follow the limitations of previous studies. Firstly, infectious diseases in the southwest of China show a seasonal pattern. Our intervention

trial will begin in April 2021 and continue for six months. Therefore, seasonal effects may not be completely eliminated during the trial period. Secondly, in the field of epidemiological trials and infection prevention, changes in subjects' behavior due to the Hawthorne effect will have a certain impact on the results. Finally, when the trial moves from the first stage to the second stage, the experimental conditions of the subjects are changed, which may affect the results due to the carry-over effect. Since the feedback intervention trial in this study is a behavioral intervention trial, it is difficult to eliminate the behavior once it is developed, so the effect cannot be eliminated by setting a washout period as commonly implemented in drug intervention trials.

## Supporting information

**S1 Appendix. SPIRIT + checklist.**
(DOCX)

**S2 Appendix. AI system in details: Deep learning and training of antibiotic prescription data in deep graph neural network technology.**
(DOCX)

## Acknowledgments

We thank all the participating institutions for providing information and assistance during the study. The authors also thank all members of the investigational team who collected the data. We acknowledge the assistance of Edward McNeil, Prince of Songkla University, Songkhla, Thailand for reading the proposal and making suggestions to improve it.

## Author Contributions

**Conceptualization:** Yue Chang, Guanghong Yang, Duan Li, Lei Wang, Lei Tang.

**Data curation:** Yue Chang, Zhezhe Cui.

**Formal analysis:** Yue Chang, Zhezhe Cui, Guanghong Yang, Duan Li, Lei Wang, Lei Tang.

**Funding acquisition:** Yue Chang.

**Investigation:** Yue Chang, Lei Wang, Lei Tang.

**Methodology:** Yue Chang, Zhezhe Cui.

**Project administration:** Yue Chang.

**Resources:** Yue Chang, Zhezhe Cui, Guanghong Yang.

**Supervision:** Zhezhe Cui, Guanghong Yang, Duan Li, Lei Wang, Lei Tang.

**Validation:** Yue Chang.

**Visualization:** Yue Chang, Duan Li, Lei Wang, Lei Tang.

**Writing – original draft:** Yuanfan Yao.

**Writing – review & editing:** Yuanfan Yao.

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
