## [Decision Letter · Decision Letter 0]

14 Jul 2021

PONE-D-21-16838

Changing antibiotic prescribing practices in outpatient primary care settings in China: study protocol for a health information system-based cluster-randomised crossover controlled trial

PLOS ONE

Dear Dr. Chang,

Thank you for submitting your manuscript to PLOS ONE. After careful consideration, we feel that it has merit but does not fully meet PLOS ONE’s publication criteria as it currently stands. Therefore, we invite you to submit a revised version of the manuscript that addresses the points raised during the review process.

We look forward to receiving your revised manuscript.

Kind regards,

Vijayaprakash Suppiah, PhD

Academic Editor

PLOS ONE

Journal Requirements:

 “The funders had and will not have a role in study design, data collection and analysis, decision to publish, or preparation of the manuscript.”

“The study was funded by the National Natural Science Foundation of China Grant on “Research on feedback intervention mode of antibiotic prescription control in primary medical institutions based on the depth graph neural network technology” (71964009) and the Science and Technology Fund Project of Guizhou Provincial Health Commission Grant on “Application Research of Deep Learning Technology in Rational Evaluation and Intervention of Antibiotic Prescription” (gzwjkj2019-1-218).”

 “The funders had and will not have a role in study design, data collection and analysis, decision to publish, or preparation of the manuscript.”

Reviewers' comments:

Reviewer's Responses to Questions

**Comments to the Author**

1. Does the manuscript provide a valid rationale for the proposed study, with clearly identified and justified research questions?

Reviewer #1: Partly

Reviewer #2: Partly

Reviewer #3: Yes

2. Is the protocol technically sound and planned in a manner that will lead to a meaningful outcome and allow testing the stated hypotheses?

Reviewer #1: Yes

Reviewer #2: Partly

Reviewer #3: Yes

3. Is the methodology feasible and described in sufficient detail to allow the work to be replicable?

Reviewer #1: Yes

Reviewer #2: No

Reviewer #3: Yes

4. Have the authors described where all data underlying the findings will be made available when the study is complete?

Reviewer #1: No

Reviewer #2: No

Reviewer #3: Yes

5. Is the manuscript presented in an intelligible fashion and written in standard English?

Reviewer #1: Yes

Reviewer #2: Yes

Reviewer #3: Yes

6. Review Comments to the Author

You may also provide optional suggestions and comments to authors that they might find helpful in planning their study.

Reviewer #1: This is an interesting protocol. The protocol used artificial intelligence - based real-time warnings, pop-up windows and educational manuals to evaluate whether those interventions could reduce antibiotic prescription rates. However, as a protocol, the article did not answer the questions whether these three can reduce the use of antibiotics. Moreover, I have some other concerns, listed as follows:

1. The introduction part needs to be more streamlined.

2. Can the results from Guizhou reflect China or other places of the world?

3. The reason for choosing Guizhou is because of the higher rate of bacterial resistance in this area? Or is the rate of overuse of antibiotics higher?

4. The study is designed to analysis whether interventions can reduce the antibiotic use rate of physicians. But it will be more interesting if the study can simultaneously analyze whether the prognosis of these doctors' patients has changed, or whether the bacterial resistance rate has changed.

5. One intervention of this protocol is "pop-up windows of antibiotic prescription rate ranking". I am curious whether this ranking will have a negative effect on medical decision-making. I mean, if in some departments where the use of antibiotics always high, such as department of infectious diseases or respiratory, will the physician change the correct medical decision because of this ranking?

6. The authors tried to analyze whether their intervention could reduce the use of antibiotics. Does the author have a theoretically reasonable interval of antibiotic usage in each department and how are they calculated? I believe that in different departments, this should be different.

7. As I mentioned before, this is a protocol. Although interesting, it still needs further expected.

Reviewer #2: In this paper, relating to the author's previous research results, doctors from township public hospitals and community health service centers in Guizhou Province were designated as participants. Two independent average formulas were used to calculate the number of doctors recruited into the study, and they were randomly divided into control groups and intervention groups. The study was divided into two stages, three months as a stage. After the completion of one stage, the two groups were crosschanged. The final evaluation indicators were a 10 day antibiotic prescription rate and an unreasonable prescription utilization rate.

The intervention model examined in this paper mainly includes the intervention early alert system of antibiotic prescription rate monitoring feedback and the artificial intelligence real-time early warning system of irrational antibiotic prescription development based on graphical neural network technology (GNN). Both of them are indicated on the doctor's computer screen in the form of an adaptive pop-up window to release early warning information. Before the preliminary study, the authors underwent a three-month pre study, and the participating institutions gave a positive answer to the intervention, believing that the intervention was beneficial to their work.

The main advantage of this paper is that the author wants to construct an artificial intelligence early detection system for antibiotic prescription control in primary medical institutions in Guizhou Province, which is technically consistent and economically feasible. Combine it with the opening of a doctor's antibiotic prescription, and reduce the antibiotic prescription rate through this intervention mode. If the intervention measures are effectively established, it will provide a good reference program for the opening of antibiotic prescriptions in the majority of hospitals, and has objective significance.

But at the moment I'm confused

1. Why do you take the way of cross cohort study?

2. What is the relationship between cross cohort and primary endpoint?

3. Because the trial scheme is open, doctors participating in the study may subjectively reduce the number of antibiotic prescriptions on the basis of understanding the content of the study. How to control the interference of this factor in this study?

Reviewer #3: This is a meaningful study. I am interested in the artificial intelligence system. Is it regional, or national? can you show me more detailed information?

7. PLOS authors have the option to publish the peer review history of their article (what does this mean?). If published, this will include your full peer review and any attached files.

Reviewer #1: No

Reviewer #2: No

Reviewer #3: No

---

## [Author Response · Author response to Decision Letter 0]

1 Aug 2021

REBUTTAL LETTER

Dear academic editor and reviewers,

Thank you very much for your review. We are grateful to the editor and 3 reviewers for giving us many good suggestions. Below are our responses. 

Journal Requirements:

Response: Thank you for the reminder. We have reviewed and revised the manuscript for formatting problems as requested by PLOS ONE. 

Response: Thank you for the reminder. We have checked all the funding information to ensure that they now match.

 “The funders had and will not have a role in study design, data collection and analysis, decision to publish, or preparation of the manuscript.”

Response: The study was funded by the National Natural Science Foundation of China Grant on “Research on feedback intervention mode of antibiotic prescription control in primary medical institutions based on the depth graph neural network technology” (71964009) and the Science and Technology Fund Project of Guizhou Provincial Health Commission Grant on “Application Research of Deep Learning Technology in Rational Evaluation and Intervention of Antibiotic Prescription” (gzwjkj2019-1-218).

Response: Our apologies. The funders had a role in the study which we should acknowledge. Specifically, all funders provided travel expenses during the data collection process, as well as the expert's expenses for providing guidance on the study design, technological guidance, and data analysis. 

Response: No author received any salary from the funders.

Response: The corresponding author, Yue Chang, was financially supported by the National Natural Science Foundation of China Grant on “Research on feedback intervention mode of antibiotic prescription control in primary medical institutions based on the depth graph neural network technology” (71964009) and the Science and Technology Fund Project of Guizhou Provincial Health Commission Grant on “Application Research of Deep Learning Technology in Rational Evaluation and Intervention of Antibiotic Prescription” (gzwjkj2019-1-218). 

Response: Yes, we have added the following to our cover letter:

“The study was funded by the National Natural Science Foundation of China Grant on “Research on feedback intervention mode of antibiotic prescription control in primary medical institutions based on the depth graph neural network technology” (71964009) and the Science and Technology Fund Project of Guizhou Provincial Health Commission Grant on “Application Research of Deep Learning Technology in Rational Evaluation and Intervention of Antibiotic Prescription” (gzwjkj2019-1-218).

The funders provided expert advice on study design, travel expenses for data collection and guidance on data analysis.

None of the authors received any salary from the funders.”

“The study was funded by the National Natural Science Foundation of China Grant on “Research on feedback intervention mode of antibiotic prescription control in primary medical institutions based on the depth graph neural network technology” (71964009) and the Science and Technology Fund Project of Guizhou Provincial Health Commission Grant on “Application Research of Deep Learning Technology in Rational Evaluation and Intervention of Antibiotic Prescription” (gzwjkj2019-1-218).”

 “The funders had and will not have a role in study design, data collection and analysis, decision to publish, or preparation of the manuscript.”

Response: Thank you. We have removed all funding-related text from the manuscript and moved it to the Funding Statement section.

The study was funded by the National Natural Science Foundation of China Grant on “Research on feedback intervention mode of antibiotic prescription control in primary medical institutions based on the depth graph neural network technology” (71964009) and the Science and Technology Fund Project of Guizhou Provincial Health Commission Grant on “Application Research of Deep Learning Technology in Rational Evaluation and Intervention of Antibiotic Prescription” (gzwjkj2019-1-218).

We have included our amended statements within the cover letter.

5. Your ethics statement should only appear in the Methods section of your manuscript. If your ethics statement is written in any section besides the Methods, please move it to the Methods section and delete it from any other section. Please ensure that your ethics statement is included in your manuscript, as the ethics statement entered the online submission form will not be published alongside your manuscript.

Response: Thank you. Our ethics statement now appears in the Methods section of the manuscript on page 23 line 389-394:

“Ethical approval and consent

The trial has been approved by the Human Trial Ethics Committee of Guizhou Medical University (Certificate No.: 2019 (148)) in Dec. 27, 2019. All the physicians at the primary care institutions participating in the trial will sign the informed consent. When collecting and analyzing data, we will remove patient confidential information, such as name and national ID. Physicians' prescription data will also be subject to strict confidentiality measures.”

Response: Thank you. We have reviewed the reference list to ensure that it is complete and correct. We did not cite any retracted article.

According to the requirements of Reviewer #1, we simplified the content of the introduction part, so the following references were deleted:

[12] Xiaoyuan Q, Yin C, Sun X, et al. Consumption of antibiotics in Chinese public general tertiary hospitals (2011-2014): Trends, pattern changes and regional differences. Plos One. 2018;13(5).

[14] Wang J, Wang P, Wang X, et al. Use and prescription of antibiotics in primary health care settings in China. Jama Intern Med. 2014;174(12):1914-1920.

[15] Radon K, Saathoff E, Pritsch M, et al. Protocol of a population-based prospective COVID-19 cohort study Munich, Germany (KoCo19). BMC Public Health. 2020;20(1):1036.

[23] Davey P, Marwick CA, Scott CL, et al. Interventions to improve antibiotic prescribing practices for hospital inpatients. Cochrane Database Syst Rev. 2017;2:CD003543.

[45] Benke K, Benke G. Artificial Intelligence and Big Data in Public Health. Int J Environ Res Public Health. 2018;15(12).

[46] Zhang X, Perez-Stable EJ, Bourne PE, et al. Big Data Science: Opportunities and Challenges to Address Minority Health and Health Disparities in the 21st Century. Ethn Dis. 2017; 27(2):95-106.

[48] Dong L, Yan H, Wang D. Antibiotic prescribing patterns in village health clinics across 10 provinces of Western China. J Antimicrob Chemother. 2008;62(2):410-415.

In addition, we updated the following 10 old and inappropriate references with references in bold: 

[7] Froom J, Culpepper L, Jacobs M, et al. Antimicrobials for acute otitis media? A review from the International Primary Care Network. BMJ. 1997;315(7100):98-102.

[7] Christaki E, Marcou M, Tofarides A. Antimicrobial Resistance in Bacteria: Mechanisms, Evolution, and Persistence. Journal of Molecular Evolution. 2020 ;88(1):26-40.

[8] Neuhauser M, Weinstein A, Rydman R, et al. Antibiotic resistance among gram-negative bacilli in US intensive care units: implications for fluoroquinolone use. JAMA. 2003; 289(7):885-888.

[8] Septimus EJ. Antimicrobial Resistance: An Antimicrobial/Diagnostic Stewardship and Infection Prevention Approach. Medical Clinics of North America. 2018 ;102(5):819-829.

[19] De Santis G, Harvey J, Howard D, et al. Improving the quality of antibiotic prescription patterns in general practice. The role of educational intervention. Med J Aust. 1994;160(8):502-505.

[15] Dekker ARJ, Verheij TJM, Broekhuizen BDL, Butler CC, Cals JWL, Francis NA, et al. Effectiveness of general practitioner online training and an information booklet for parents on antibiotic prescribing for children with respiratory tract infection in primary care: a cluster randomized controlled trial. Journal of Antimicrobial Chemotherapy. 2018 ;73(5):1416-1422.

[20] Briel M, Langewitz W, Tschudi P, et al. Communication training and antibiotic use in acute respiratory tract infections. A cluster-randomised controlled trial in general practice. Swiss Med Wkly. 2006;136(15-16):241-247.

[16] Little P, Stuart B, Francis N, Douglas E, Tonkin-Crine S, Anthierens S, et al. Antibiotic Prescribing for Acute Respiratory Tract Infections 12 Months After Communication and CRP Training: A Randomized Trial. Annals of Family Medicine. 2019;17(2):125-132.

[31] Welschen I, Kuyvenhoven MM, Hoes AW, et al. Effectiveness of a multifaceted intervention to reduce antibiotic prescribing for respiratory tract symptoms in primary care: randomised controlled trial. BMJ. 2004;329(7463):431.

[26] Pettersson E, Vernby A, Mölstad S, Lundborg CS. Can a multifaceted educational intervention targeting both nurses and physicians change the prescribing of antibiotics to nursing home residents? A cluster randomized controlled trial. Journal of Antimicrobial Chemotherapy. 2011;66(11):2659-2666.

[34] Bjerrum L, Cots JM, Llor C, et al. Effect of intervention promoting a reduction in antibiotic prescribing by improvement of diagnostic procedures: a prospective, before and after study in general practice. European Journal of Clinical Pharmacology. 2006;62(11):913.

[29] Gerber JS, Prasad PA, Fiks AG, Localio AR, Grundmeier RW, Bell LM, et al. Effect of an outpatient antimicrobial stewardship intervention on broad-spectrum antibiotic prescribing by primary care pediatricians: a randomized trial. JAMA. 2013;309(22):2345-2352.

[35] Altiner A, Brockmann S, Sielk M, et al. Reducing antibiotic prescriptions for acute cough by motivating GPs to change their attitudes to communication and empowering patients: a cluster-randomized intervention study. J Antimicrob Chemother. 2007;60(3):638-644.

 [30] Butler CC, Simpson SA, Dunstan F, Rollnick S, Cohen D, Gillespie D, et al. Effectiveness of multifaceted educational programme to reduce antibiotic dispensing in primary care: practice based randomised controlled trial. BMJ. 2012;(344):d8173.

[56] LaValley, Michael P. Intent-to-treat Analysis of Randomized Clinical Trials. ACR/ARHP Annual Scientific Meeting 2003.

[49] McCoy CE. Understanding the Intention-to-treat Principle in Randomized Controlled Trials. Western Journal of Emergency Medicine. 2017;18(6):1075-1078.

[61] Hongzhou S, Qinjian Y, Qianjin Z. Review of Development and Application of Delphi Method in China — One of Series Papers of Nanjing University Knowledge Mapping Research Group. Journal of Modern Information. 2011;31(5).

[54] Araújo V, Teixeira PM, Yaphe J, Correia de Sousa J. The respiratory research agenda in primary care in Portugal: a Delphi study. BMC Family Practice. 2016 ;17(1):124. 

[62] Yingyao C, Ming N, Yaozhi H, et al. Selection of indicators to measure public interest of public medical institutions: Based on the Delphi method. Chinese Journal of Health Policy. 2012;5(1):6-10. 

[55] Banno M, Tsujimoto Y, Kataoka Y. The majority of reporting guidelines are not developed with the Delphi method: a systematic review of reporting guidelines. Journal of Clinical Epidemiology. 2020; 124:50-57.

Reviewer #1

This is an interesting protocol. The protocol used artificial intelligence - based real-time warnings, pop-up windows and educational manuals to evaluate whether those interventions could reduce antibiotic prescription rates. However, as a protocol, the article did not answer the questions whether these three can reduce the use of antibiotics. Moreover, I have some other concerns, listed as follows:

(1) The introduction part needs to be more streamlined.

Response: According to your suggestion, we have shortened the introduction on page 4-7, line 51-125:

“Antibiotic resistance is a widespread concern around the world. In the past decade, 50% of antibiotic prescriptions have been used to treat colds, and, according to a report of World Health Organization (WHO), many of these cases have no indication of antibiotic use. The main forms of antibiotic overuse and abuse are non-indication, overdose, and multi-drug combined use of antibiotics, which violate the principles of antibiotic use. Overuse and abuse of antibiotics are major risk factors for antibiotic resistance. The total consumption of antibiotics in 71 countries (including China) increased by more than 50% from 2000 to 2010. The antibiotic prescription rate in China is twice as high as that recommended by WHO, and higher than that in developed countries and most developing countries. Guizhou province, located in southwest China, has the largest number of poor people and is the largest poverty-stricken area in China. Our retrospective study of Guizhou's 39 primary care facilities showed that most of the medical staff do not hold a university degree, and the rate of inappropriate antibiotic prescriptions was over 90%. The incidence of bacterial resistance in Guizhou province is on the rise.

Previous studies have shown that there are a variety of methods for antibiotics prescription control, including educational intervention, communication training, nursing point testing, electronic decision support system, and delayed prescription, but reports of a feedback intervention are rare. Existing feedback interventions mainly focused on email or poster information, regular or irregular assessment/audit of antibiotic prescriptions, or prescription recommendations from experts and peers delivered at a meeting or online. Some studies even report prescribing information publicly. However, these methods are somewhat mandatory and censored, which can cause negative emotions to the physician, and needs long-term intervention by professionals. As a result, some studies have shown negative results. 

We previously conducted a cluster randomised crossover-controlled trial to reduce antibiotic prescription rates based on an existing health information system (HIS) in primary care institutions in Guizhou. In this study, the antibiotic prescription rates decreased by 15% over the 6-month study period. However, a limitation was that prescription of unreasonable antibiotics were not take into account. In view of this, this new study will analyze and process the digital information in electronic medical records with big data using depth map neural network (DMNN) in artificial intelligence (AI) technology to provide physicians with the best diagnosis and treatment suggestions in real time. 

This is a cluster-randomised, open controlled, crossover, superiority trial. We aim to describe an automatically-presented, privacy-protecting, DMNN technology-based feedback intervention model. The feedback intervention model can not only effectively remind physicians of the deviation of their prescribing behavior from their peers, but also humanely give reasonable suggestions, which can greatly improve the enthusiasm of physicians to participate. In addition, an educational handbook developed by us for primary outpatient institutions will be distributed to all primary physicians. A pilot study will be conducted to test the motivation of the physicians and effectiveness of the intervention. The comparators are usual care, i.e. primary care hospitals within Guizhou which did not receive any intervention.”

(2) Can the results from Guizhou reflect China or other places of the world?

Response: Antibiotic abuse varies widely from region to region in China. The economic level of Guizhou province is low and the health resources are scarce. Due to the limited medical capacity of physicians in primary care institutions, antibiotic abuse is relatively serious in Guizhou (Chang Y, 2019). Therefore, the results of this study are intended to be used in areas with high levels of antibiotic abuse, rather than nationwide. At the same time, we expanded the sample size based on the original study (Chang Y, 2020), and also increased the reliability of this research results.

We have added a sentence in the discussion section reflecting these ideas. Page 23, line 404-405:

“Based on the results of our preliminary study, this intervention is suitable for primary care institutions in developing countries with paperless office conditions.”

(3) The reason for choosing Guizhou is because of the higher rate of bacterial resistance in this area? Or is the rate of overuse of antibiotics higher?

Response: Guizhou is one of China's poorest provinces. The level of medical care is relatively low. Our previous study (Chang Y, 2019) showed that the rate of overuse and misuse of antibiotic prescriptions in primary care institutions in Guizhou was over 90%. Therefore, Guizhou is very representative. In addition, the incidence of bacterial resistance in Guizhou province is on the rise.

In the background section on lines 77-83 on page 5, we focused on explaining why Guizhou Province was chosen as the research site:

“Guizhou province, located in southwest China, has the largest number of poor people and is the largest poverty-stricken area in China. Our retrospective study of Guizhou’s 39 primary care facilities showed that most of the medical staff have less than a university education, and the unreasonable rate of antibiotic prescriptions was over 90%. The incidence of bacterial resistance in Guizhou province is on the rise.”

(4) The study is designed to analysis whether interventions can reduce the antibiotic use rate of physicians. But it will be more interesting if the study can simultaneously analyze whether the prognosis of these doctors' patients has changed, or whether the bacterial resistance rate has changed.

Response: Thank you for your advice, which is exactly the direction of our future research. In addtion, as we mentioned in the outcome measurement section on page 17-18, our goal is not only to reduce the prescribing rate of antibiotics (primary outcome), but also to reduce the irrational prescribing rate of antibiotics (secondary outcome). We have also added a sentence on page 3, line 40: 

“…to effectively reduce antibiotic prescription rates and antibiotic irregularities.”

(5) One intervention of this protocol is "pop-up windows of antibiotic prescription rate ranking". I am curious whether this ranking will have a negative effect on medical decision-making. I mean, if in some departments where the use of antibiotics always high, such as department of infectious diseases or respiratory, will the physician change the correct medical decision because of this ranking?

Response: Thank you. This comment is thought-provoking. In our previous study (Yue Chang, 2019/2020), we found that the primary care institutions in Guizhou Province were mainly general clinics, and a few large hospitals had independent clinics of traditional Chinese medicine, orthopedics and gynecology. Considering the feasibility of our outpatient ranking intervention, this study only included GPs in the general practice room. In addition, we also highlighted the situation of GPs in the "recruitment section" of our methodology on page 9 line 165:

“ …2) each primary care institution has at least 3 outpatient general physicians (GPs), …”

Furthermore, we also pointed out in the "Intervention Section" :

“The information seen by each physician will be confidential. The physicians have the freedom to read this feedback message or not.”

We think this is the mildest and most compliant type of feedback intervention. In conclusion, we therefore believe that, in principle, the ranking system will not interfere with the physician’s decision to prescibe the correct medication, rather, it will prompt those who have a high rate of prescribing, relative to their peers, to reconsider their behaviour carefully. 

(6) The authors tried to analyze whether their intervention could reduce the use of antibiotics. Does the author have a theoretically reasonable interval of antibiotic usage in each department and how are they calculated? I believe that in different departments, this should be different.

Response: As mentioned above, we are dealing with primary care physicians who are treating patients with common diseases at the primary care level. According to our previous study (Chang Y, 2020), 10 days was the most reasonable interval. The inclusion criteria for physicians was a minimum of 100 prescriptions in 10 days. This facilitated our calculation of antibiotic prescription rates. The transition model we use is also well suited for this kind of crossover design study with the same interval. Our ranking feedback intervention is updated every 10 days, three times a month. Our previous Chinese article collected the opinions of physicians, who also thought that three times a month was a reasonable rate. So, we didn't design different time intervals for the general outpatients. 

(7) As I mentioned before, this is a protocol. Although interesting, it still needs further expected.

Response: Thank you very much for your comments.

Reviewer #2

(1) Why do you take the way of cross cohort study?

Response: A crossover design study is a repeated measurement study in which each unit receives different treatments at different times. A crossover design study is commonly considered in RCT because it is more effective than a parallel design. The crossover design not only allows all participants to experience all interventions, but can also use inter-group comparisons to reduce confounding factors (Berggren L, 2021; Senn S, 2002; David H DS, 2016). We also mentioned this on page 9 line 177-178：

“To avoid selection bias, we will use a crossover design in which each group will receive different treatments at different times.”

(2) What is the relationship between cross cohort and primary endpoint?

Response: Firstly, as mentioned on page 11, lines 183-186,

“The trial will be divided into two stages and will last for 6 months. The first stage will last for 3 months, with Group 1 enrolled in the intervention group and Group 2 in the control group. The second stage will also last for 3 months, with the two groups crossing over.”

We will compare the antibiotic prescription rates from baseline to the cross-over point (after 3 months) and from the cross-over point to the end point (after 6 months) in Group 1 and Group 2, respectively. The antibiotic prescription rates in Group 1 and Group 2 were also compared at baseline (0 month), cross-over point (3 months), and end point (6 months). 

We also mentioned this on page 19 line 329-331:

“We will compare the antibiotic prescription rate and the antibiotic prescription rationality rate between the two groups (Group 1 and Group 2) at baseline, cross-over point and at the end of the trial.”

(3) Because the trial scheme is open, doctors participating in the study may subjectively reduce the number of antibiotic prescriptions on the basis of understanding the content of the study. How to control the interference of this factor in this study?

Response: Unlike drug intervention trials, our feedback interventions were designed to improve the prescribing behavior of the doctors. This type of behavioral intervention is inherently difficult to define when to start and when to end, which is why our cross-over design has no washout period. At the same time, due to the good effect achieved in the early intervention studies (Chang Y, 2020), we have also received the support of Guizhou Provincial Health Commission and will carry out long-term prescription intervention in primary health hospitals in the future. 

So, we expect that doctors will automatically reduce their prescription of antibiotics and increase the rationality of antibiotics because of our intervention, which is our long-term goal. 

Reviewer #3

(1) This is a meaningful study. I am interested in the artificial intelligence system. Is it regional, or national? can you show me more detailed information?

Response: Thank you for your query. This was a regional study. In the future, we also want to scale it up to the whole country and even to other developing countries. We will provide more details about the AI system in the Appendix (Appendix AI system in details). 

Kind Regards.

Yuanfan Yao

07-31-2021

Reference

[1] Chang Y, Sangthong R, McNeil EB, et al. Effect of a computer network-based feedback program on antibiotic prescription rates of primary care physicians: A cluster randomized crossover-controlled trial. J Infect Public Health. 2020.

[2] Chang Y, Chusri S, Sangthong R, et al. Clinical pattern of antibiotic overuse and misuse in primary healthcare hospitals in the southwest of China. PLoS One. 2019;14(6):e0214779.

[3] Berggren L. Analysing a cross-over study. Statistical work and challenges related to planning, conducting and analysing a clinical trial with cross-over design. Sweden: Stockholm University; 2012.

[4] Senn S. Cross-over trials in clinical research (second edition). Department of statistical scientice and department of epidemiology and public health university college London, UK: John Wiley & sons, Ltd.; 2002.

[5] David H DS, Eduardo B. Which Treatment Is Better? Ascertaining Patient Preferences with Crossover Randomized Controlled Trials. J Pain Symptom Manage. 2016;49(3): 625–631. doi:10.1016/j.jpainsymman.2014.11.294.

---

## [Decision Letter · Decision Letter 1]

12 Oct 2021

Changing antibiotic prescribing practices in outpatient primary care settings in China: study protocol for a health information system-based cluster-randomised crossover controlled trial

PONE-D-21-16838R1

Dear Dr. Chang,

We’re pleased to inform you that your manuscript has been judged scientifically suitable for publication and will be formally accepted for publication once it meets all outstanding technical requirements.

Kind regards,

Vijayaprakash Suppiah, PhD

Academic Editor

PLOS ONE

Reviewer's Responses to Questions

**Comments to the Author**

1. Does the manuscript provide a valid rationale for the proposed study, with clearly identified and justified research questions?

Reviewer #1: Yes

Reviewer #2: Yes

2. Is the protocol technically sound and planned in a manner that will lead to a meaningful outcome and allow testing the stated hypotheses?

Reviewer #1: Partly

Reviewer #2: Yes

3. Is the methodology feasible and described in sufficient detail to allow the work to be replicable?

Reviewer #1: Yes

Reviewer #2: Yes

4. Have the authors described where all data underlying the findings will be made available when the study is complete?

Reviewer #1: Yes

Reviewer #2: Yes

5. Is the manuscript presented in an intelligible fashion and written in standard English?

Reviewer #1: Yes

Reviewer #2: Yes

6. Review Comments to the Author

You may also provide optional suggestions and comments to authors that they might find helpful in planning their study.

Reviewer #1: The authors have answered well and revised the manuscript. This manuscript is ready to publish and I have no other concerns.

Reviewer #2: The author kindly answered our questions, clearly explained why the cross cohort study was adopted, and explained the relationship between cohort design and expected endpoint. However, how to reduce the subjective reduction of antibiotic prescriptions when participating doctors know the research content still needs to further design better research methods to reduce this interference factor.At the same time, the real data have not been obtained, so it is difficult to evaluate the specific effect of this method.

7. PLOS authors have the option to publish the peer review history of their article (what does this mean?). If published, this will include your full peer review and any attached files.

Reviewer #1: No

Reviewer #2: No

---

## [Editor Report · Acceptance letter]

15 Oct 2021

PONE-D-21-16838R1 

Changing antibiotic prescribing practices in outpatient primary care settings in China: study protocol for a health information system-based cluster-randomised crossover controlled trial 

Dear Dr. Chang:

I'm pleased to inform you that your manuscript has been deemed suitable for publication in PLOS ONE. Congratulations! Your manuscript is now with our production department. 

Kind regards, 

on behalf of

Dr. Vijayaprakash Suppiah 

Academic Editor

PLOS ONE